# Exploring Figure-Ground Assignment Mechanism in Perceptual Organization

**Wei Zhai**[1]  **Yang Cao**[1,3*]  **Jing Zhang**[2]  **Zheng-Jun Zha**[1]

[1]University of Science and Technology of China  [2]The University of Sydney
[3]Institute of Artificial Intelligence, Hefei Comprehensive National Science Center
{wzhai056,forrest,zhazj}@ustc.edu.cn, jing.zhang1@sydney.edu.au

## Abstract

Perceptual organization is a challenging visual task that aims to perceive and group the individual visual element so that it is easy to understand the meaning of the scene as a whole. Most recent methods building upon advanced Convolutional Neural Network (CNN) come from learning discriminative representation and modeling context hierarchically. However, when the visual appearance difference between foreground and background is obscure, the performance of existing methods degrades significantly due to the visual ambiguity in the discrimination process. In this paper, we argue that the figure-ground assignment mechanism, which conforms to human vision cognitive theory, can be explored to empower CNN to achieve a robust perceptual organization despite visual ambiguity. Specifically, we present a novel Figure-Ground-Aided (FGA) module to learn the configural statistics of the visual scene and leverage it for the reduction of visual ambiguity. Particularly, we demonstrate the benefit of using stronger supervisory signals by teaching (FGA) module to perceive configural cues, *i.e.*, convexity and lower region, that human deem important for the perceptual organization. Furthermore, an Interactive Enhancement Module (IEM) is devised to leverage such configural priors to assist representation learning, thereby achieving robust perception organization with complex visual ambiguities. In addition, a well-founded visual segregation test is designed to validate the capability of the proposed FGA mechanism explicitly. Comprehensive evaluation results demonstrate our proposed FGA mechanism can effectively enhance the capability of perception organization on various baseline models. Nevertheless, the model augmented via our proposed FGA mechanism also outperforms state-of-the-art approaches on four challenging real-world applications.

## 1  Introduction

Perceptual organization, which is a vital visual task, refers to the processes by which the disjoint bits of visual information are structured into the larger coherent units that we eventually experience as environmental objects. Thanks to the developments of discriminative representation learning and hierarchical context modeling with convolutional neural networks [17, 33, 48, 77], the past few years have witnessed tremendous progress in perceptual organization. Recent advanced methods are usually based on the framework of fully convolutional network (FCN) [5, 33, 50, 76], which learn a discriminative feature representation and hierarchically model the local context by supervision derived from human-given labels.

However, the performance of existing methods degrades significantly when deployed in some challenging tasks with complex visual ambiguity, such as camouflaged object detection [11, 27],

---

*Corresponding author.

36th Conference on Neural Information Processing Systems (NeurIPS 2022).

medical image segmentation [12, 50], and visual industrial detection [74]. Since the visual appearance differences between the foreground and background are obscure, it is difficult to perceive the correlation between individual visual elements and determine the boundaries. The visual ambiguities impede CNN's represent learning and contextual modeling, leading to inaccurate and incomplete perceptual organization.

To address this issue, more perceptual knowledge is required to be incorporated into visual organization. In the 1920s, the Gestalt psychologists identified Grouping and Figure-Ground as two important principles underlying the process of perceptual organization [24, 25, 53, 66]. Grouping principle refers to bring together individual visual elements that produce stimuli to form a holistic perception. It has attracted more attention because it is intuitively similar to the contextual modeling ability of CNN. Different from Grouping, Figure-Ground assignment refers to the perception that assigns a boundary separating two regions to one of them [52]. By introducing the configural statistics of the natural world in which the visual system evolved [3], the Figure-Ground assignment principle is thought to be

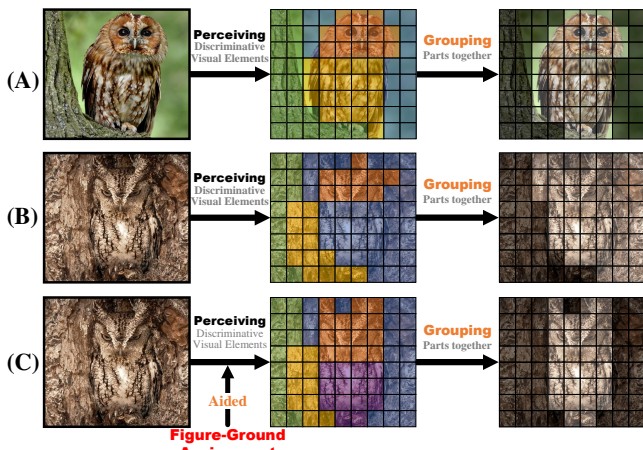

Figure 1: **Main idea.** (A) The process of a perceptual organization first requires perceiving discriminative visual elements, then segregation is obtained by grouping parts together. (B) This process is obscured in scenarios with ambiguity. (C) In this paper, we explore Figure-Ground Assignment to facilitate in reducing visual ambiguities. Best viewed in color.

important in reducing the visual ambiguousness of a scene. In addition, Cognitive science studies [15, 47, 81] have found that neural activity associated with the Figure-Ground assignment mechanism in the V2 cortex of human vision occurs as early as 10-25 ms after the generation of visual stimuli, providing strong support for the role of local bottom-up processing. Another study [43] on the "meaningfulness principle" also showed that assigned figures tend to be associated with neighborhoods with familiar shapes, pointing to the integration of knowledge from the top-down.

Inspired by these studies, as shown in Fig. 1, we argue that exploring figure-ground assignment mechanism can empower CNN the ability of perceptual organization despite visual ambiguity, and consequently presents a novel Figure-Ground-Aided (FGA) module that learns the configural statistics of visual scene, and leverages it for representation learning. Specifically, we firstly investigate the configural cues related to the Figure-Ground assignment mechanism in human psychophysics and find that the figural region usually takes on the shape instructed by the separating boundary and appears closer to the viewer, while the ground region is seen as extending behind the figure [43]. Typically, as shown in Fig. 2 (a), Convexity cue [21, 34, 39, 44] corresponds to the regions on either side of the boundary where the scene depth changes abruptly and is beneficial for analyzing the hierarchical relationship between neighboring regions in the image, facilitating hierarchical contextual modeling. And lower region cues [16, 62] usually corresponds to the region of the scene where occlusion has occurred and is beneficial for analyzing the occlusion relationship between various neighborhoods in an image and determining the shape attribution of foreground objects and background regions.

After that, instead of using a weak form of directly using mask labels as supervision, we refer to the method in [32] of teaching the FGA module to perceive the configural cues that human deem important for the perceptual organization by using human-derived labels as stronger supervisory signals. Notably, the Ground-Truth (GT) segmentation masks imply the prior knowledge of the annotators' understanding of natural scenes. Therefore, we easily generate the labels of the two configural cues from the Ground-Truth (GT) label without the intervention of additional information and exploit them as supervisions for the FGA module, facilitating the modeling of the configural statistics of natural scenes. Furthermore, an Interactive Enhancement Module (IEM) is presented to progressively enhance the discriminative features for boundary assignment via a local/global interactive strategy. Specifically, a collaborative local interaction process is first introduced by swapping queries to align locally between contextual co-occurring configural features. Then, a global

interaction process is introduced to establish global spatial correlations of local configural features, resulting in complete object boundaries. Moreover, a Lambda strategy is utilized to improve the computational efficiency and performance of the original self-attention module.

To systematically investigate the performance of our proposed method for the Figure-Ground assignment, we design a synthetic computer vision task inspired by an important experiment in cognitive science—Figure-Ground Segregation [26, 46, 49, 55]. Furthermore, the comprehensive experiments demonstrate that our proposed FGA module can facilitate the CNN to learn more efficiently in the reduction of visual ambiguities with low data requirements. Finally, we also validate the performance of our proposed mechanism in four challenging visual applications, including camouflaged object detection [11, 71], polyp detection [12], and lung anomaly detection [13]. The results demonstrate the superiority of our method over SOTA methods. The contributions are summarized as follows:

(1) This paper explores the Figure-Ground assignment mechanism from human vision cognitive theory to empower CNN to learn configural statistics to reduce visual ambiguities, thereby achieving robust perceptual organization. (2) This paper presents a novel Figure-Ground-Aided module to integrate figure-ground-aided cues in a hierarchical manner. An Interactive Enhance-

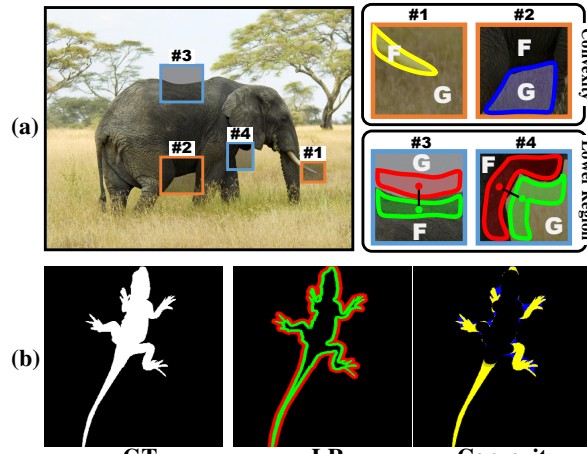

Figure 2: **Figure-Ground Cues.** (a) The Figure (F)-Ground (G) assignment cues used in this paper. We implement the two cues of Convexity and Lower Region as two 2-channel labels, which are directly calculated from GT labels without the intervention of additional information. We represent the concave region and the convex region as different labels to reflect their structural differences for the convexity cues. We represent intra-boundary and extra-boundary regions differently for lower region cues to reflect the differences between the two sides of the boundary. (b) The example of cues. "Convexity" represents the convexity cue. "LR" represents the lower region cue. Best viewed in color.

ment Module is devised to progressively enhance the discriminative features for Figure-Ground assignment via a local/global interactive strategy. (3) This paper introduces the Figure-Ground Segregation test, a synthetic visual perception challenge to systematically investigate the performance of models in Figure-Ground assignment. Experimental results demonstrate our proposed FGA module consistently improves the Figure-Ground assignment performance of several representative networks on datasets of different difficulty levels. (4) Extensive experiments are also performed on four challenging robust object segmentation applications, showing that the model constructed via our FGA module outperforms SOTAs.

## 2   Methodology

**Architecture.** In this paper, we propose a novel Figure-Ground-Aided (FGA) module that uses configuration cues from the Figure-Ground assignment process to enhance the perception of foreground-background relationship, thereby achieving a robust perceptual organization result. It can be easily incorporated into existing encoder-decoder models and improve their performance. Fig. 3 shows the overall architecture of the proposed Figure-Ground-Aided module. Our proposed FGA module is composed of two cue-aided branches and an Interactive Enhancement Module (IEM). Here, we use ResNet-50 [17] as the backbone. Specifically, we remove the fully connected layer and retain all convolutional bocks. Given an input image of shape $H \times W$, this backbone will generate five scales of features with spatial resolution gradually decreasing by stride 2. We denote these features as $\mathcal{E} = \{E_k \mid k = 1, 2, 3, 4, 5\}$. The size of the $k$-th feature is $\left[C_k, \frac{H}{2^k}, \frac{W}{2^k}\right]$, where $C_k$ is the number of channels of the $k$-th feature. We utilize the feature from $\{E_k \mid k = 1, 2, 3, 4\}$ as skip-connected feature for FGA module and the decoder. $\mathcal{D} = \{D_k \mid k = 1, 2, 3, 4, 5\}$ are the feature maps of the decoder part. The intermediate features $E_k^{Con}$ and $E_k^{LR}$ are derived from the feature $E_5$. The calculation process here is the same as in the Decoder part.

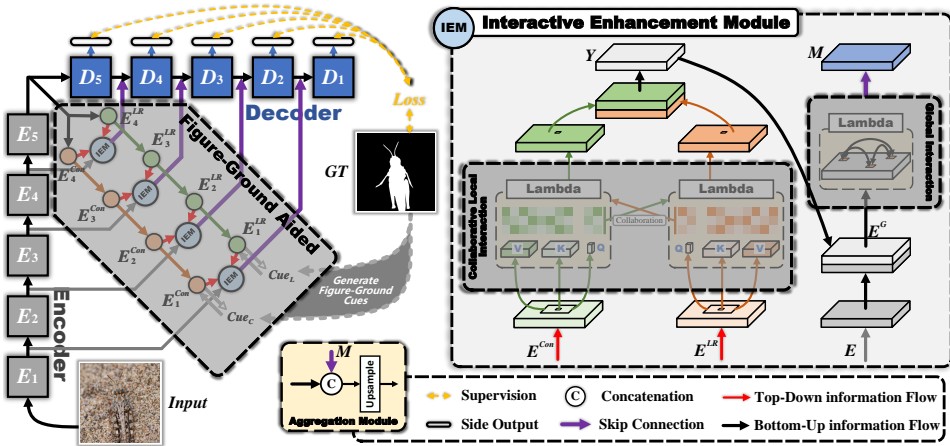

Figure 3: **The architecture of the proposed Figure-Ground Aided module (FGA).** Figure-Ground Aided module, which can be easily incorporated into encoder-decoder models, is composed of two cue-aided branches and an Interactive Enhancement Module (IEM). The IEM is proposed to integrate the features at different scales of branches separately to enhance the feature of the decoder branch. We leverage the convexity and lower region cues, which can be calculated directly from the GT label and do not require any additional information, as supervisions for perceiving the contextual information of object boundary, respectively. We first allow the network to learn the statistical features related to the configuration in a top-down manner. Then we use the learned features to guide the bottom-up features from the encoder to integrate bottom-up and top-down information. **Interactive Enhancement Module (IEM)**, which consists of two parts: Collaborative Local Interaction (CLI) and Global Interaction (GI). IEM has three input features ($E^{\text{Con}}$, $E^{\text{LR}}$, and $E$), two of which come from the intermediate feature learned by the Figure-Ground assignment cues aided branches ($E^{\text{Con}}$ and $E^{\text{LR}}$). "Lambda" indicates the use of a Lambda strategy [1] to improve the computation efficiency of the CLI and GI interaction.

**Figure-Ground Assignment Cues.** The critical process of the perceptual organization known as figure-ground assignment [51], involves giving one of the two adjacent regions a boundary. The figure-Ground assignment is commonly thought to follow region segmentation, and it is an essential step in forming a perception of surfaces, shapes, and objects [63, 64]. The human visual mechanism points out that when humans observe images, they will use some configural cues to distinguish between foreground and background, including Convexity [21, 34, 39, 44] and Lower region [16, 62]. Accordingly, we propose to introduce the convexity and lower region cues, as shown in Fig. 2 (b). To generate the convexity cue, we used morphological opening and closing operations [4, 45]. It can be formed as: $Cue_C = \texttt{Cat}(Cue_{C1}, Cue_{C2})$, "$\texttt{Cat}$" is the concatenation operation.

$$Cue_{C1} = GT - \Phi_\circ(GT, K_C), \tag{1}$$

$$Cue_{C2} = \Phi_\bullet(GT, K_C) - GT, \tag{2}$$

where $GT$ is the ground truth segmentation mask. $\Phi_\circ(.)$ is the opening operation. $\Phi_\bullet(.)$ is the closing operation. $K_C$ is the structure element whose size is 10. We get the convexity Cue by concatenating $Cue_{C1}$ and $Cue_{C2}$, whose shape is $[2, H, W]$. $H$ and $W$ are the height and width of the mask. To generate the lower region cue, we used morphological erosion and expansion operations. It can be expressed as: $Cue_L = \texttt{Cat}(Cue_{L1}, Cue_{L2})$.

$$Cue_{L1} = GT - \Phi_\ominus(GT, K_L), \tag{3}$$

$$Cue_{L2} = \Phi_\oplus(GT, K_L) - GT, \tag{4}$$

where $\Phi_\ominus(.)$ is an erosion operation. $\Phi_\oplus(.)$ is the expansion operation. $K_L$ is the structure element whose size is 5. We get lower region cue by concatenating $Cue_{L1}$ and $Cue_{L2}$.

**Interactive Enhancement Module.** To effectively utilize the important configural cues (convexity and lower region) provided by the Figure-Ground assignment mechanism, a novel Interactive Enhancement Module (IEM) is devised. The IEM has three input features, two of which come from the intermediate feature learned by the Figure-Ground assignment cue-aided branches. We define them

as $E^{\text{Con}}$ and $E^{\text{LR}}$, respectively. Moreover, the remaining branch is the skip-connection feature $E$ from the encoder. We further improve the representation ability of the encoder feature by establishing the interactions among the features of the skip-connection features and aided branches feature. As shown in Fig. 3, the interactions in the IEM consist of collaborative local interaction and global interaction. 1) Collaborative Local (CLI) interaction: the local interaction of the two Figure-Ground cues integrates the local structural context features. 2) Global interaction (GI): the global interaction between base features and local contextual features aims to enhance foreground features and suppress the background with structural context features.

The detailed structure of the Interactive Enhancement Module is illustrated in Fig. 3. We use a spatial self-attention mechanism that considers both content and location interactions to achieve local or global interaction of features. This paper uses an efficient Lambda layer [1], which captures interactions of feature elements by transforming available contexts into linear functions, termed lambdas, and applying these linear functions to each input separately. For the collaborative local interaction part, it can be expressed as follows:

$$Y = \texttt{Lambda}(E^{\text{LR}}, P^{\text{LR}})Q^{\text{Con}} + \texttt{Lambda}(E^{\text{Con}}, P^{\text{Con}})Q^{\text{LR}}, \tag{5}$$

where $Y$ is the output of CLI. $P$ denotes the relative position embeddings. $Q^{\text{LR}}$ is the query, which is computed by: $Q^{\text{LR}} = E^{\text{LR}}W_Q^{\text{LR}}$. $Q^{\text{Con}}$ is calculated in the same way. $W_Q^{\text{LR}}$ is the learnable weights. $\texttt{Lambda}(.)$ is defined as:

$$\texttt{Lambda}(E^{\text{Cue}}, P^{\text{Cue}}) = (\texttt{Softmax}(E^{\text{Cue}}W_K^{\text{Cue}}))^{\top}(E^{\text{Cue}}W_V^{\text{Cue}}) + (P^{\text{Cue}})^{\top}(E^{\text{Cue}}W_V^{\text{Cue}}), \tag{6}$$

where the range of relative position in CLI is set to 5. $\text{Cue} \in LR, Con$ For the global interaction part, it can be expressed as follows:

$$M = \texttt{Lambda}(E^{\text{G}}, P^{\text{G}})Q^{\text{G}}, \tag{7}$$

where $E^G = \texttt{Cat}(Y, E)$. $E^G$ and $M$ correspond to the feature maps shown on the right side of the IEM in Fig. 3. The $\texttt{Lambda}(.)$ function is defined as follows:

$$\texttt{Lambda}(E^{\text{G}}, P^{\text{G}}) = (\texttt{Softmax}(E^{\text{G}}W_K^{\text{G}}))^{\top}(E^{\text{G}}W_V^{\text{G}}) + (P^{\text{G}})^{\top}(E^{\text{G}}W_V^{\text{G}}), \tag{8}$$

where the range of relative position in GI is set to the whole feature map size.

**Loss Function.** In this section, we introduce the loss function used to train FGA-Net. We adopt a deep supervision strategy for each sub-side output prediction map from the decoder and calculate the binary cross-entropy (BCE) [40] loss. The total loss $\mathcal{L}_{\text{D}}$ for the decoder of the proposed network could be expressed as follows: $\mathcal{L}_{\text{D}} = \sum_i^I \mathcal{L}_d^{(i)}$, where $\mathcal{L}_d^{(i)}$ represents the loss of the $i$-th sub-side output prediction. For the other two aided branches in FGA module, we also use BCE loss to obtain the losses $\mathcal{L}_{\text{Con}}$ and $\mathcal{L}_{\text{LR}}$. Unlike the decoder, we do not use a deep supervision strategy. The final loss for the entire network is defined as follows: $\mathcal{L}_{\text{total}} = a\mathcal{L}_{\text{D}} + b\mathcal{L}_{\text{Con}} + c\mathcal{L}_{\text{LR}}$, where $a$, $b$, and $c$ are the hyper-parameters to balance different loss terms. In this paper, we set $a = 1$, $b = 0.9$, and $c = 0.9$, respectively. Note that all experiments use the same hyper-parameters.

## 3 Experiments

### 3.1 Figure-Ground Segregation Test

To investigate the validity of our proposed Figure-Ground Aided module (FGA), we design and establish a set of tasks to evaluate the Figure-Ground assignment ability of deep convolutional neural networks, inspired by the Figure-Ground Segregation test in cognitive science experiments [26, 46, 49, 55] as shown in Fig. 4 (a). This task is described in detail as shown in Fig. 4 (b), by sampling the content of a randomly given texture image and filling the foreground and background of the given image using that texture content to generate the desired sample finally.

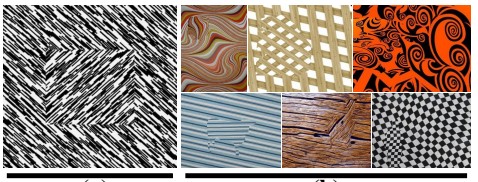

(a)        (b)

Figure 4: **Figure-Ground Segregation Test.** (a) Exemplars from the Figure-Ground Segregation experiment in cognitive research. (b) Exemplars from the FGS test.

Here we use a synthetic dataset to test a deep convolutional neural network to remove distracting

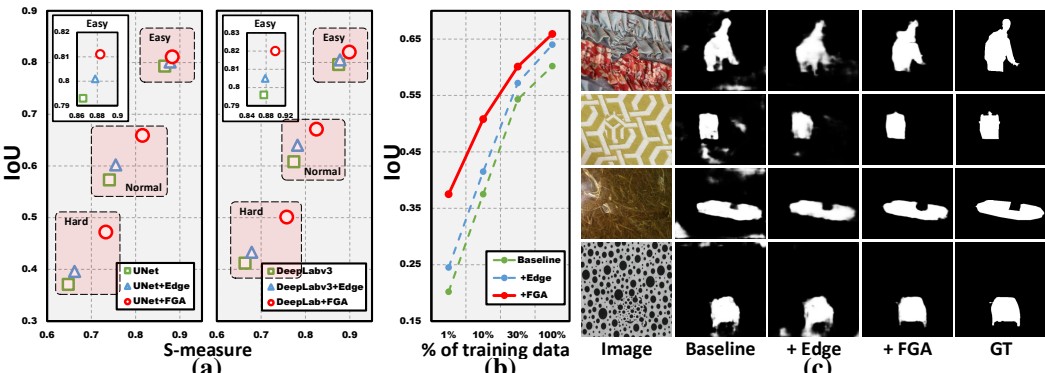

Figure 5: **FGA module has a better capability of Figure-Ground assignment.** (a) FGA module consistently improves the Figure-Ground assignment performance of several representative networks on all three datasets. Different dash boxes represent different levels. The x-axis and the y-axis are the S-measure and IoU scores, respectively. UNet+Edge/DeepLabv3+Edge indicates UNet/DeepLabv3 with auxiliary edge supervision. UNet+FGA/DeepLabv3+FGA means UNet/DeepLabv3 with FGA module. (b) FGA-Net is more data-efficient. The x-axis and the y-axis represent different proportions of the normal dataset and the IoU score, respectively. (c) Visual comparisons on the Normal level dataset. The UNet [50] is used as our baseline model.

factors. Standard computer vision datasets make it difficult to pinpoint the relative contributions of different visual strategies since the performance of architecture may be affected by several factors, including dataset biases, model hyper-parameters, and the number of samples [22, 31].

**Test Design.** We use segmentation labels from the Pascal VOC [8] dataset and the rich texture dataset (DTD [7]) to synthesize our dataset. Referring to the cognitive science experiments which exclude irrelevant factors from the test conditions, such as excluding other irrelevant factors for Figure-Ground Segregation, we also need to exclude the influence of several factors, *i.e.*, 1) excluding the influence of contextual information brought by semantic labels since utilizing semantic labels of different objects helps to learn discriminative feature representation; 2) excluding the influence of spatial contexts provided by multiple instances belonging to the same class since they contribute to learning robust (*e.g.*, scale-invariant or occlusion-aware) feature representation; and 3) excluding segmentation cues due to significant appearance differences between foreground and background since their distinct textures [72, 73], colors, and illumination help to learn cheap features to distinguish them. Guided by the above principles, the whole process of sample generation can be divided into the following steps: 1) select a random image in the Pascal dataset and use one of the object instances as a figure and the remaining regions as the ground; 2) given a collection of texture images, randomly select a texture image from it; 3) two random transformations (such as rotation and scaling) are performed on the texture image independently; and 4) fill the figure region and the ground region with the two transformed textures respectively, resulting in a synthetic sample. In addition, for a more comprehensive evaluation, three datasets of different difficulty levels (Easy, Normal, and Hard) are established by varying the transformations, the division of the set of texture images, and the size of figure regions. Each dataset contains 2,500 unique images with a 224 × 224 resolution, split into training (2,000) and test (500) sets.

**Experimental Conditions.** In this paper, we conduct experiment on two representative architectures: U-Net [50] and Deeplabv3 [6]. U-Net is one of the most popular architectures in the field of segmentation, which has a typical encoder-decoder architecture, and the shallow features are directly fed into the decoder through skip connections. Deeplabv3 is a highly competitive network for semantic segmentation. Through atrous spatial pyramid pooling, it can expand the receptive field while ensuring detailed modeling. Each model is trained using the Adam optimizer with a batch size of 16 and a learning rate of $1e^{-4}$ for the Figure-Ground Segregation test. To isolate the influence of pre-trained weights, all weights are randomly initialized. We train each model for 20,000 iterations. For the evaluation of the results, we focus on the differences between the output and the ground truth in terms of structure and filling completeness, so we choose S-measure [9], and IoU [33] as evaluation metrics.

| | LR | Covx | IEM | $S_\alpha$ | IoU |
|---|:---:|:---:|:---:|:---:|:---:|
| Baseline | | | | .745 | .580 |
| (a) | ✓ | | | .771 | .612 |
| (b) | | ✓ | | .761 | .603 |
| (c) | ✓ | ✓ | | .793 | .635 |
| FGA | ✓ | ✓ | ✓ | .816 | .659 |

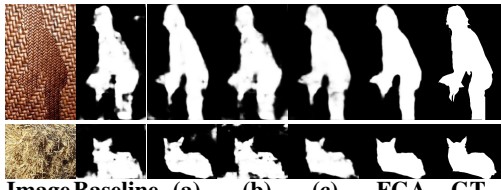

Image Baseline (a) (b) (c) FGA GT

Figure 6: **Ablation study of the proposed FGA-Net on the normal dataset.** "Covx" ("LR") denotes convexity (lower region). The UNet [50] is used as our baseline model.

**Result and Discussion.** Through experiments on the Figure-Ground segregation test, we can get the following four conclusions. ▶**First**, with different difficulty levels, the proposed FGA can consistently improve the performance of the base model (in Fig. 5 (a)). It shows that our method has a strong capability of the Figure-Ground assignment by introducing human cognitive cues. ▶**Second**, as shown in Fig. 5 (a), our proposed FGA-Net has a better performance compared to the straightforward strategy of directly adding edge supervision. Its segmentation results are more complete in terms of both structure and filling of foreground regions as shown in Fig. 5 (c). Using the edge supervision can only provide local perception at the foreground boundary, while more structural priors and perception of the foreground region are required for the Figure-Ground assignment. ▶**Third**, as shown in Fig. 5 (b), it can be seen that as the number of training samples decreases, our model still outperforms the contenders. Our FGA trained on only $1\%$ of the dataset is comparable to Baseline+Edge trained on $10\%$ of the dataset, suggesting that FGA is more data-efficient. The supervisions from the Figure-Ground cues help the network learn an inductive bias [68] that the network focuses on the foreground and background and their differences at the boundaries, thereby reducing the amount of data required. ▶**Fourth**, as shown in Fig. 6, the LR cue provides a more complete edge for segmentation results, the Convexity cue enhances the perception of shape detail, and the IEM enhances the consistency within the foreground. In addition, we find that using two branches to learn both priors separately is more effective than using a single branch to learn both priors. This phenomenon explains to some extent the certain orthogonality between the two cues, which depend on different features. And we also conduct an ablation study on the IEM, as shown in Table 1, which identifies the performance contribution of each sub-module in the IEM. It is worth mentioning that the local interactions bring more improvement than the global interactions. This may be due to the boundary assignment process that tends to capture discriminative features from the local region.

Table 1: **Ablation study of IEM on the normal dataset.** "Local" means use local interaction. "Collaborative" means use collaborative strategey. "Global" means use global interaction. "Lambda" indicates the use of Lambda [1] strategy.

| Local | Collaborative | Global | Lambda | $S_\alpha$ | IoU |
|:---:|:---:|:---:|:---:|:---:|:---:|
| ✓ | | | | .801 | .640 |
| ✓ | ✓ | | | .805 | .645 |
| | | ✓ | | .797 | .638 |
| ✓ | ✓ | ✓ | | .810 | .652 |
| ✓ | ✓ | ✓ | ✓ | .816 | .659 |

## 3.2 Applications

We verify the effectiveness of our proposed method on four challenging visual tasks, *i.e.*, Camouflaged Object Detection (*COD*) [11], Polyp Segmentation (*PS*) [12], and Lung Infection Segmentation (*LIS*) [13]. The COD task mainly manifests in complex background/edges, small targets/structures, slender trunks/limbs, and partially occluded objects. The main feature of the PS task is its high diversity of size and texture. The main difficulty of the LIS task lies in the complex structures of the foreground. In addition, we also verify the effectiveness of our proposed method on the non-challenging figure-ground task.

**Camouflaged Object Detection. Datasets:** We evaluate our model on three challenging visual Camouflaged Object Detection datasets, *i.e.*, CHAMELEON (**CHA**) [59], CAMO (**CAM**) [27], and COD10K (**COD**) [11]. CHA is collected from the Internet by searching "camouflaged animal" as the keyword. CAM is the first publicly released COD dataset. COD is the most challenging dataset with higher image quality. **Implementation Details:** We implement our model with PyTorch, and TITAN Xp GPUs are used for training and testing. ResNet-50 [17] is used as a encoder in our model, which is initialized as the pre-trained weights on ImageNet. In this paper, we use the resolution of $480 \times 480$ during both the training phase and testing phase. Our model is trained for 100 epochs using the Adam

Table 2: **Comparison with** 6 **SOTA methods on the CHA [59], CAM [27], and COD [11] datasets.** ↑ indicates higher is better.

| | | [78] | [11] | [10] | [70] | [38] | [69] | Ours |
|---|---|---|---|---|---|---|---|---|
| CHA [59] | $S(\uparrow)$ | .848 | .869 | .888 | .893 | .882 | .888 | **.902** |
| | $E(\uparrow)$ | .870 | .891 | .942 | .923 | .942 | .918 | **.947** |
| | $F(\uparrow)$ | .702 | .740 | .816 | .813 | .810 | .796 | **.840** |
| | $M(\downarrow)$ | .050 | .044 | **.030** | **.030** | .033 | .031 | **.030** |
| CAM [27] | $S(\uparrow)$ | .732 | .751 | **.820** | .775 | .782 | .785 | .803 |
| | $E(\uparrow)$ | .768 | .771 | **.882** | .847 | .852 | .859 | .871 |
| | $F(\uparrow)$ | .583 | .606 | .743 | .673 | .695 | .686 | **.748** |
| | $M(\downarrow)$ | .104 | .100 | .070 | .088 | .085 | .086 | **.068** |
| COD [11] | $S(\uparrow)$ | .727 | .771 | .815 | .814 | .800 | .818 | **.821** |
| | $E(\uparrow)$ | .779 | .806 | .887 | .865 | .868 | .850 | **.895** |
| | $F(\uparrow)$ | .509 | .551 | .680 | .666 | .660 | .667 | **.687** |
| | $M(\downarrow)$ | .056 | .051 | .037 | .035 | .040 | .035 | **.031** |

Table 3: **Comparison with six SOTA methods on the COVID-19 CT segmentation dataset.**

| | Dice(↑) | Sen.(↑) | Spec.(↑) | $S(\uparrow)$ | $E(\uparrow)$ | $M(\downarrow)$ |
|---|---|---|---|---|---|---|
| [50] | .439 | .534 | .858 | .622 | .625 | .186 |
| [41] | .583 | .637 | .921 | .744 | .625 | .112 |
| [54] | .623 | .658 | .926 | .725 | .739 | .102 |
| [29] | .515 | .594 | .840 | .655 | .814 | .184 |
| [82] | .581 | .672 | .902 | .722 | .662 | .120 |
| [13] | .682 | .692 | .943 | .781 | .720 | .082 |
| [20] | .700 | **.751** | – | – | .860 | .084 |
| Ours | **.754** | .748 | **.973** | **.799** | **.911** | **.056** |

Table 4: **Performance on DUTS-Test [65] and PASCAL-S [30].**

| | DUTS-Test | | | | PASCAL-S | | | |
|---|---|---|---|---|---|---|---|---|
| | $M(\downarrow)$ | $F(\uparrow)$ | $S(\uparrow)$ | $E(\uparrow)$ | $M(\downarrow)$ | $F(\uparrow)$ | $S(\uparrow)$ | $E(\uparrow)$ |
| [79] | .041 | .807 | .885 | .914 | .062 | .800 | .858 | .891 |
| [67] | .035 | .840 | .892 | .927 | .062 | .825 | .862 | .901 |
| [28] | **.032** | .866 | .899 | .937 | **.061** | .824 | .863 | .903 |
| Ours | .033 | **.868** | **.902** | **.940** | **.061** | **.827** | **.866** | **.907** |

Table 5: **Comparison with four SOTA methods on Kvasir, CVC-612, ColonDB, ETIS, and Endo datasets.**

| | | [50] | [82] | [14] | [12] | [80] | Ours |
|---|---|---|---|---|---|---|---|
| Kvasir [18] | Dice(↑) | .818 | .821 | .723 | .898 | .907 | **.911** |
| | IoU(↑) | .746 | .743 | .611 | .840 | **.862** | .858 |
| | $F(\uparrow)$ | .794 | .808 | .670 | .885 | .893 | **.898** |
| | $S(\uparrow)$ | .858 | .862 | .782 | .915 | .922 | **.922** |
| | $E^m(\uparrow)$ | .893 | .910 | .849 | .948 | .944 | **.953** |
| | $M(\downarrow)$ | .055 | .048 | .075 | .030 | .028 | **.025** |
| CVC-612 [2] | Dice(↑) | .823 | .794 | .700 | .899 | .921 | **.924** |
| | IoU(↑) | .755 | .729 | .607 | .849 | .879 | **.884** |
| | $F(\uparrow)$ | .811 | .785 | .647 | .896 | .914 | **.930** |
| | $S(\uparrow)$ | .889 | .873 | .793 | .936 | .941 | **.943** |
| | $E^m(\uparrow)$ | .954 | .931 | .885 | .979 | .972 | **.982** |
| | $M(\downarrow)$ | .019 | .022 | .042 | .009 | **.008** | **.008** |
| ColonDB [60] | Dice(↑) | .512 | .483 | .469 | .709 | .755 | **.768** |
| | IoU(↑) | .444 | .410 | .347 | .640 | .678 | **.683** |
| | $F(\uparrow)$ | .498 | .467 | .379 | .696 | .737 | **.746** |
| | $S(\uparrow)$ | .712 | .691 | .634 | .819 | .836 | **.842** |
| | $E^m(\uparrow)$ | .776 | .760 | .765 | .869 | **.883** | .868 |
| | $M(\downarrow)$ | .061 | .064 | .094 | .045 | .041 | **.040** |
| ETIS [58] | Dice(↑) | .398 | .401 | .297 | .628 | .719 | **.723** |
| | IoU(↑) | .335 | .344 | .217 | .567 | **.664** | .651 |
| | $F(\uparrow)$ | .366 | .390 | .231 | .600 | .678 | **.680** |
| | $S(\uparrow)$ | .684 | .683 | .557 | .794 | **.840** | .822 |
| | $E^m(\uparrow)$ | .740 | .776 | .633 | .841 | .830 | **.834** |
| | $M(\downarrow)$ | .036 | .035 | .109 | .031 | .020 | **.015** |
| Endo [61] | Dice(↑) | .710 | .707 | .467 | .871 | .869 | **.889** |
| | IoU(↑) | .627 | .624 | .329 | .797 | .807 | **.817** |
| | $F(\uparrow)$ | .684 | .687 | .341 | .843 | .849 | **.865** |
| | $S(\uparrow)$ | .843 | .839 | .640 | .925 | .925 | **.929** |
| | $E^m(\uparrow)$ | .876 | .898 | .817 | .972 | .943 | **.978** |
| | $M(\downarrow)$ | .022 | .018 | .065 | .010 | .010 | **.007** |

[23] optimizer with an initial learning rate of 0.0001, decreased by 0.1 at 50 epochs. The batch size is 32. **Evaluation Criteria:** S-measure ($S_\alpha$) [9], mean E-measure ($E_\phi$) [37], weighted F-measure ($F_\beta^\omega$) [37], and Mean Absolute Error (*MAE*) [42] are used as the evaluation metrics. **Comparison with SOTA:** We compare the performance of FGA-Net with SOTA methods, including **EGNet** [78], **SINet** [11], **SINetv2** [10], **MGL** [70], **PFNet** [38], and **UGTR** [69]. Quantitative results are listed in Table 2. Obviously, our model outperforms the contenders. In particular, FGA-Net achieves significant performance improvement compared to the second-best method SINetv2 [10] on the challenging COD10K dataset. It proves that with the help of Figure-Ground assignment mechanism, it can make the network more efficient in solving the problem of separating ambiguous regions, which is caused by camouflaged objects that are highly similar to background in appearance.

**Polyp Segmentation. Datasets:** We evaluate our model on five challenging Polyp Segmentation (*PS*) datasets, Kvasir (**Kvasir**) [18], CVC-ClinicDB/CVC-612 (**CVC-612**) [2], CVC-ColonDB (**ColonDB**) [60], ETIS (**ETIS**) [58], and EndoScene (**Endo**) [61]. Kvasir is the most challenging dataset, which contains 1,000 images. CVC-612 dataset includes 612 open-access images from 31 colonoscopy clips. ColonDB and ETIS consist of 380 and 196 polyp images. Note that the training set only contains the data of Kvasir and CVC-612. We follow [12] to split all datasets. **Implementation Details:** We use the same implementation method as COD task, except that the batch size is 16. We use the resolution of $352 \times 352$ during both the training phase and testing phase. **Evaluation Criteria:** Dice [19], IoU [18], $F_\beta^\omega$, $S_\alpha$, maximum E-measure ($E_\phi^{max}$), and *MAE* are used as the metrics. **Comparison with SOTA:** We compare our FGA-Net with four SOTA medical image segmentation methods, including **UNet** [50], **UNet++** [82], **SFA** [14], **Pra** [12], and **MSNet**

[80]. Quantitative results are listed in Table 5. We can observe that our FGA-Net surpasses the aforementioned methods in most of the metrics. Compared to the second-best methods on each dataset, FGA-Net outperforms all of them on five datasets. Similar results can also be observed in terms of the MAE score.

**Lung Infection Segmentation. Datasets:** We also evaluate our model for Lung Infection Segmentation (*LIS*) on the COVID-19 CT segmentation dataset, including 100 axial CT images from different COVID-19 patients. We follow [13] to randomly select 45 CT images as training samples, 5 images for validation, and 50 images for testing. **Implementation details:** We use the same implementation method as COD task, except that the batch size is 24. We use the resolution of $352 \times 352$ during both the training phase and testing phase. **Evaluation Criteria:** Dice, Sensitivity (Sen.) [56], Specificity (Spec.) [57], $S_\alpha$, $E_\phi^{mean}$, and *MAE* are used as the metrics. **Comparison with SOTA:** We compare our FGA-Net with six SOTA medical image segmentation methods including **UNet** [50], **UN++** [82], **AUN** [41], **GUN** [54], **DUN** [29], **Inf** [13], and **ERRNet** [20]. Quantitative results are listed in Table 3. We can observe that our model consistently outperforms all other contenders across all metrics.

**Non-Challenging Figure-Ground Task.** In Table 4, we conduct experiments on more non-challenging Figure-Ground datasets (salient object detection), and the results show that the FGA-Net proposed in our paper is also competent for salient object detection. It indicates that FGA helps for different segmentation tasks by improving the boundary assignment capability of the model, which in turn improves the final performance.

# 4 Related Work

Historically, the visual phenomenon most closely associated with the perceptual organization is grouping, and Figure-Ground assignment [24, 25, 53, 66]. In general, grouping determines what the qualitative elements of perception are, and figure-ground assignment determines the interpretation of those elements in terms of their shapes and relative locations in the layout of surfaces in the real-world [63, 64]. Figure-Ground assignment refers to the perceptual process of assigning a boundary separating two regions to one of them [52] and provides important configural prior for scene perception. Accordingly, the studies of the Figure-Ground assignment process [43, 63, 64] show that it plays a central role in aiding higher levels of visual perception.

Therefore, we hope to introduce the Figure-Ground assignment process into the segmentation model to help address the object segmentation. Factors that affect Figure-Ground assignment include size, surroundedness, orientation and contrast [52], symmetry [15], parallelism [39], convexity [21, 34, 39, 44], meaningfulness [43], and lower region [62]. In this paper, we mainly consider the cues related to the configuration, *i.e.*, convexity and lower region. Compared to recent research [35, 36] applying the Figure-Ground Assignment mechanism in some vision tasks without considering high-level supervisions, our work instead explores an end-to-end figure-ground aided approach that can be easily incorporated into existing encoder-decoder models and improve their performance.

# 5 Conclusion

In this paper, we demonstrate that the Figure-Ground cues inspired by the perceptual organization of human vision can be effectively utilized to improve the performance of a CNN model for different challenging robust perception organization tasks, *e.g.*, camouflaged object detection, polyp segmentation, and lung infection segmentation. Specifically, we investigate the proposed FGA module, on a carefully established synthetic dataset for the Figure-Ground Segregation test, which aims to measure the capability of Figure-Ground Assignment. The empirical study shows that the proposed new module consistently improves the performance of several representative networks in reducing visual ambiguities. Moreover, the model implemented via our proposed FGA mechanism outperforms SOTA approaches in three challenging real-world applications. We believe this study provides valuable insights to the community and attracts attention to exploring the Figure-Ground principles for other visual perception tasks.

**Societal Impact.** Perceptual organization is one of the most challenging computer vision tasks. Albeit being challenging, Perceptual organization is beneficial to a wide range of applications [75, 83]. However, there also exists the risk that the technology is utilized in the scenario of the illegal shoot, malicious edit, and incorrect use.

**Acknowledgement** This work was supported by National Key R&D Program of China under Grant 2020AAA0105701, National Natural Science Foundation of China (NSFC) under Grants 62225207, 61872327 and U19B2038, and the University Synergy Innovation Program of Anhui Province under Grants GXXT-2019-025.

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
