# OpenReview forum: "Exploring Figure-Ground Assignment Mechanism in Perceptual Organization"
_NeurIPS.cc/2022/Conference — NeurIPS 2022 Accept_

### Official Review · Reviewer_cfAi · 2022-07-08

**Rating:** 6
**Confidence:** 4
**Soundness:** 3 good
**Presentation:** 3 good
**Contribution:** 3 good

**Summary:**

This work proposes a representation learning method to achieve the Perceptual Organization process implicitly. The authors design a new module named as Figure-Ground-Aided (FGA), in which the convexity and lower region cues are leveraged by the Interactive Enhancement Module (IEM). The model takes traditional CNNs as the backbone. The optimization utilizes segmentation labels and binary cross-entropy (BCE) loss. In the optimization, the convexity and lower region cues derived by segmentation labels are also used as the supervision. Experiments based on the synthetic and realistic datasets are designed to verify the effectiveness. Overall, the motivation of this paper is clear and the designed model is technically sound. Moreover, some claims of this paper are not factual.

**Questions:**

1. Why authors design the experiments on the synthetic dataset? If the realistic datasets may include some other irrelevant factors for Figure-Ground Segregation, authors should explain this problem explicitly.
2. There are some mistakes in the notations and expressions:
(1) In Eq. (7), the notation L_M has no explanation.
(2) In line 277, authors state that “We verify the effectiveness of our proposed method on four challenging visual tasks”. However, the following statement only mention three tasks (COD, PS, LIS).
(3) In Figure 7, authors state that “Comparison with 10 SOTA methods ...”. However, only six comparison methods are involved.
(4) In the second line of Figure 10, the best result is not 0.858 but 0.862. The bold style is marked incorrectly. Authors should check this kind of mistake carefully.
(5) In Figure 9, the results of [66] on two different datasets are identical. Please check it.
(6) In line 148, the subscript i has no explanation.
(7) In line 193, the notations “i" and “I” has no explanation.
(8) Figures 7, 8, 9 and 10 should be marked as tables.


**Limitations:**

The authors give the societal impact in Section 6. However, there is no limitations discussed in this section. The reviewer believes that the requirement of segmentation labels should be a limitation. Specifically, the traditional object detection models only require box labels, which is cheaper than segmentation labels.

**Strengths And Weaknesses:**

Strengths:
1. The originality is good. This work is inspired by cognitive researches. The motivation is clear and interesting, while the designed model is novel and technically sound.
2. The paper is organized well and the statements are clear.
3. The significance is enough. This work aims to learn visual representations being discriminative to the confusing objects and backgrounds. This is an essential problem in computer vision.

Weaknesses:
1. Some necessary implementation details are missing:
   (1) The model architecture is not introduced completely. In Figure 3, the intermediate features E^Con_4, ..., E^Con_1 and E^LR_4, ..., E^LR_1 are derived from the feature E_5. However, there is no description for this process.
   (2) The size of structure element K_C is 10 while that of K_L is 5. Why authors select these hyperparameters?
   (3) In the ablation experiments, it is not clear how to implement the variant models (a), (b) and (c) without the IEM module.

2. In the main paper, there is no direct evidence verifying that the proposed model can reducing the visual ambiguousness in realistic datasets. Although a large number of realistic datasets are used in the comparison experiments, only the overall quantitative results are shown in the paper. The improvements of these quantitative results may be unrelated to the reduction of visual ambiguousness.
  Therefore, the ablation experiments on the realistic datasets are also necessary. Note that the experimental results on the synthetic datasets cannot be considered as a direct evidence for this problem.

3. Some claims of this paper are not factual.
  (1) In line 297, authors claim that “FGA-Net achieves significant performance improvement compared to the second-best method UGTR [67]”. However, UGTR [67] should not be the second-best method according to the results in Figure 7. For example, SINetv2 [10] is better than UGTR [67] for most of the metrics. Moreover, the superiority of the proposed model is not very significant compared with SINetv2 [10].
  (2) In line 325, authors claim that “Quantitative results are listed in Table 8. We can observe that our model consistently outperforms all other contenders across all metrics”. However, the proposed model is inferior than [20] for the metric Sen.

---

> ### Author Response · Authors · 2022-08-02
> **To Reviewer cfAi (1/2)**
>
> We thank the reviewer for taking the time to read our responses and provide positive assessments and additional concerns. We wish to have the opportunity to address the concerns regarding empirical evaluations.
>
> **Q#:** Why authors design the experiments on the synthetic dataset? If the realistic datasets may include some other irrelevant factors for Figure-Ground Segregation, authors should explain this problem explicitly.
>
> **A#:** As a cognitive visual testing, Figure-Ground Segregation is presented in Section 3. Similar to the control variable method in physical experiments, Figure-Ground Segregation can reliably evaluate the capability of object segmentation models in boundary assignment, excluding irrelevant factors and some short-cut cues (e.g., context information). Specifically, referring to the cognitive science experiments which exclude irrelevant factors from the test conditions, such as excluding other irrelevant factors for Figure-Ground Segregation, we also need to exclude the influence of several factors, i.e. 1) excluding the influence of contextual information brought by semantic labels since utilizing semantic labels of different objects helps to learn discriminative feature representation; 2) excluding the influence of spatial contexts provided by multiple instances belonging to the same class since they contribute to learning robust (e.g., scale-invariant or occlusion-aware) feature representation; and 3) excluding segmentation cues due to significant appearance differences between foreground and background since their distinct textures, colors, and illumination help to learn cheap features to distinguish them.
>
> From the perspective of computer vision, the competition of this test is some kind of weak. However, this is precisely the characteristic of cognitive testing. Standard computer vision datasets make it difficult to pinpoint the relative contributions of different visual strategies since the performance of architecture may be affected by several factors, including dataset biases, model hyper-parameters, and the number of samples [ref1][ref2][ref3][ref4].
>
> [ref1] D. Linsley, J. Kim, V. Veerabadran, and T. Serre. Learning long-range spatial dependencies with horizontal gated-recurrent units. NeurIPS, 2018.
>
> [ref2] J. Kim, D. Linsley, K. Thakkar, and T. Serre. Disentangling neural mechanisms for perceptual grouping. ICLR, 2020.
>
> [ref3] Linsley, D., Malik, G., Kim, J., Govindarajan, L.N., Mingolla, E. and Serre, T.. Tracking Without Re-recognition in Humans and Machines. NeurIPS, 2021.
>
> [ref4] Vaishnav, M., Cadene, R., Alamia, A., Linsley, D., VanRullen, R. and Serre, T., 2022. Understanding the computational demands underlying visual reasoning. Neural Computation.
>
>
> **Q#:** The ablation experiments on the realistic datasets are also necessary.
>
> **A#:** Following the reviewer’s advice, we added the relevant ablation experiments on the realistic dataset, as shown in the following table, and the performance is generally consistent with that in the Figure-Ground Segregation Test. Thanks for your suggestion, and due to space limitations in the main paper, we have included these ablation results in the supplementary material (Table S6 and Table S7).
>
> Ablation study of the proposed FGA-Net on COD dataset.
> |             |  LR       | Covx      | IEM      | S      | E      | F      | M      |
> |:--------:|:--------:|:--------:|:--------:|:--------:|:--------:|:--------:|:--------:|
> | Baseline |           |           |           |  0.795  |  0.878  |  0.649  |  0.063  |
> | (a)      |  $\surd$  |           |           |  0.808  |  0.871  |  0.669  |  0.040  |
> | (b)      |           |  $\surd$  |           |  0.803  |  0.866  |  0.654  |  0.039  |
> | (c)      |  $\surd$  |  $\surd$  |           |  0.813  |  0.880  |  0.683  |  0.035  |
> | FGA      |  $\surd$  |  $\surd$  |  $\surd$  |  0.821  |  0.895  |  0.687  |  0.031  |
>
> Ablation study of IEM on COD dataset.
> |    Local    |  Collaborative  |  Global  |  Lambda  | S      | E      | F      | M      |
> |:--------:|:--------:|:--------:|:--------:|:--------:|:--------:|:--------:|:--------:|
> | $\surd$ |           |           |           |  0.815  |  0.884  |  0.680  |  0.037  |
> | $\surd$ |  $\surd$  |           |           |  0.816  |  0.889  |  0.690  |  0.036  |
> |         |           |  $\surd$  |           |  0.813  |  0.880  |  0.681  |  0.037  |
> | $\surd$ |  $\surd$  |  $\surd$  |           |  0.818  |  0.890  |  0.686  |  0.033  |
> | $\surd$ |  $\surd$  |  $\surd$  |  $\surd$  |  0.821  |  0.895  |  0.687  |  0.031  |

---

> > ### Author Response · Authors · 2022-08-02
> > **To Reviewer cfAi (2/2)**
> >
> > **Q#:** The superiority of the proposed model is not very significant
> >
> > **A#:** Our work aims to explore the F-G assignment mechanism to empower CNN to achieve a robust boundary assignment despite visual ambiguity. And we present a novel Figure-Ground-Aided (FGA) module to learn the configural statistics of the visual scene and leverage it for the reduction of visual ambiguity. To clearly discuss its effectiveness, a cognitive visual testing--Figure-Ground Segregation is presented in Section 3. Similar to the control variable method in physical experiments, Figure-Ground Segregation can reliably evaluate the capability of object segmentation models in boundary assignment, excluding irrelevant factors and some short-cut cues (e.g., context information). The comprehensive experiments adequately demonstrate that our proposed FGA module can facilitate the CNN to learn more efficiently in the reduction of visual ambiguities with low data requirements.
> >
> > Furthermore, the application tasks serve as validation of the potential of our approach. Experimental results show that we achieve superior results in most metrics for all four applications (COD, PS, LIS, and SOD), and we believe that careful tuning of the parameters will result in better performance. To demonstrate the potential of our method, we have optimized some hyperparameters (learning rate and decay strategy) based on the original model, and the experiments show that the performance of our method can be further improved. Where "Our*" represents the performance after optimizing the hyperparameters.
> >
> > The results on CHA Dataset.
> > |            |  S | E      | F      | M      |
> > |:--------:|:--------:|:--------:|:--------:|:--------:|
> > |   [10]   |  0.888   | 0.942  | 0.816  | 0.030  |
> > |   [67]   |  0.888   | 0.945  | 0.796  | 0.031  |
> > |    Our      |  0.898   | 0.945  | 0.839  | 0.032  |
> > |    Our*     |  0.902   | 0.949  | 0.840  | 0.031  |
> >
> > The results on CAM Dataset.
> > |             |  S       | E      | F      | M      |
> > |:--------:|:--------:|:--------:|:--------:|:--------:|
> > |   [10]   |  0.820   | 0.882  | 0.743  | 0.070  |
> > |   [67]   |  0.785   | 0.859  | 0.686  | 0.086  |
> > |    Our      |  0.793   | 0.858  | 0.745  | 0.078  |
> > |    Our*     |  0.803   | 0.871  | 0.748  | 0.068  |
> >
> > The results on COD Dataset.
> > |             |  S       | E      | F      | M      |
> > |:--------:|:--------:|:--------:|:--------:|:--------:|
> > |   [10]   |  0.815   | 0.887  | 0.680  | 0.037  |
> > |   [67]   |  0.818   | 0.850  | 0.667  | 0.035  |
> > |    Our      |  0.818   | 0.888  | 0.683  | 0.034  |
> > |    Our*     |  0.821   | 0.895  | 0.687  | 0.031  |
> >
> > The results on COVID-19 Dataset.
> > |             |  Dice       | Sen      | Spec      | S      | E      | M      |
> > |:--------:|:--------:|:--------:|:--------:|:--------:|:--------:|:--------:|
> > |   [13]   |  0.682   | 0.692  | 0.943  | 0.781  | 0.720  | 0.082  |
> > |   [20]   |  0.700   | 0.751  | -  | -  | 0.860  | 0.084  |
> > |    Our      |  0.735   | 0.720  | 0.965  | 0.792  | 0.900  | 0.062  |
> > |    Our*     |  0.754   | 0.748  | 0.973  | 0.799  | 0.911  | 0.056  |
> >
> > [10] D.-P. Fan, G.-P. Ji, M.-M. Cheng, and L. Shao. Concealed object detection. T-PAMI, 2021.
> >
> > [13] D.-P. Fan, T. Zhou, G.-P. Ji, Y. Zhou, G. Chen, H. Fu, J. Shen, and L. Shao. Inf-net: Automatic covid-19 lung infection segmentation from ct images. T-MI, 2020.
> >
> > [20] G.-P. Ji, L. Zhu, M. Zhuge, and K. Fu. Fast camouflaged object detection via edge-based reversible re-calibration network. PR, 2021.
> >
> > [67] F. Yang, Q. Zhai, X. Li, R. Huang, A. Luo, H. Cheng, and D.-P. Fan. Uncertainty-guided transformer reasoning for camouflaged object detection. CVPR, 2021.
> >
> >
> > **Q#:** Some necessary implementation details are missing.
> >
> > **A#:** (1) The intermediate features E^Con_4, ..., E^Con_1 and E^LR_4, ..., E^LR_1 are derived from the feature E_5. The calculation process here is the same as in the Decoder part. (2) The structure element size mentioned in the paper is an intuitive choice. As shown in Section 4, our proposed method achieves competitive results without hyper-parameter tuning. We believe that a careful selection of this hyperparameter will improve performance. (3) We have presented the relevant details in supplementary materials to facilitate a better understanding of the reader (Figure S6 in the supplementary material).
> >
> >
> > **Q#:** There are some mistakes in the notations and expressions.
> >
> > **A#:** (1): We are sorry to miswrite "L_D" as "L_M". (2): The four tasks here include, in addition to the previously mentioned COD, PS, and LIS, the SOD task mentioned at the end of the experimental section. (5): We are sorry to miswrite. The results of [66] on PASCAL-S dataset is 0.062, 0.825, 0.862, and 0.901. (3)(4)(6)(7)(8): Thank you for the reminder. We have checked and fixed these issues as suggested. Please refer to our rebuttal revision.

---

### Official Review · Reviewer_ghhU · 2022-07-09

**Rating:** 4
**Confidence:** 5
**Soundness:** 2 fair
**Presentation:** 3 good
**Contribution:** 2 fair

**Summary:**

The paper proposes a Figure-Ground-Aided (FGA) module to improve the performance of existing CNN models in figure-ground segregation. In particular, the method considers two figure-ground assignment cues, i.e., convexity and lower-region, and derives two supervisory signals automatically from ground-truths for network supervision. An interactive enhancement module is proposed to enhance feature representations. The results look good.

**Questions:**

- I am particularly concerning  whether the lower-region cue is well connected to the challenge studied in the work. More discussions are demanded. In addition, I am wondering how will the model perform by directly using a edge-aware loss like HED [ref3] instead of $\mathcal{L}_{LR}$.

- I am unsure about the meaning of "stronger supervisory signals" in Line 89. Why is it a weak form by "directly using mask labels as supervision"? This is not convinced to me since many competitors in Tables 7-10 only use ground-truths as supervision, and still obtains very promising performance.

- The statement "figure-ground assignment ... contributes almost all perception-based tasks" seems an overclaim, or more discussions should be provided to support the argument.

- So many errors in only seven formulas:
  - Eq.1 and Eq.2 are not well written. What does the symbol $\Leftarrow$ refer to?
  - It is unclear the meaning fo $GT_{L_2}$ in Eq. 2.
  - How are $K_c$ and $K_L$ determined? Do they set to the same values across the datasets? How will the different values affect the performance?
  - L184 only gives the definition of $Q^{LR}$, however, strictly speaking, $Q^{Con}$ is not defined.
  - It is not clear what the $E$ in Line 187 refers to.
  - $\mathcal{L}_M$ is not defined. Should it be $\mathcal{L}_D$ in Line 193?
  - It is also not clear to me how $M$ in Eq. 5 will be used in the network.
  -  The $\texttt{Lambda}$ in Eq. 3 is very similar to a self-attention operator. Why is $\texttt{Lambda}$ used here?
  - The symbols $Cue_{LR}$ and $Cue_{Con}$ shown in Fig. 3 are not consistent with symbols in Section 2.
  - $\{D_i\}_{i=1}^5$ in Fig.3 are also not defined.
  - What does the $P_d$ refer to? The symbols is potentially misused since it is defined to denote position embedding in Line 183.

- Typos or grammar errors:
  - "fully connection layer" should be "fully connected layer"
  - "k-th feature" should be "the k-th feature"

[ref3] Xie, Saining, and Zhuowen Tu. "Holistically-nested edge detection." ICCV 2015.



**Limitations:**

No limitation is discussed.

**Strengths And Weaknesses:**

Strengths

This work is on addressing figure-ground segregation of situations with high visual ambiguity. The paper overall is well organized and easy to read in most sections (not in the technical part). Experiments on a synthetic benchmark and four practical applications are surely extensive and the results look good to me.

Weaknesses

While it makes sense to me to formulate perceptual theory into neural networks, the motivation to use the specific two cues (convexity and lower-region) instead of many other cues (size, surroundedness, orientation and contrast, symmetry, parallelism, meaningfulness) as mentioned in Related Work (Line 343) is not quite clear to me. The two cues explored here have also been explored previously, e.g., in [ref1,ref2], thus the novelty seems to be limited. Regarding the performance, even though that the authors argue that "existing methods degrades significantly when deployed in cases with complex visual ambiguity", I found from Tables 7-10 that the improvements are very minor in most datasets. Thus, I am concerning the core motivation of the method and the effectiveness of the technical implementations. Last, the formulations in Section 2 are not well presented. Though I can get the main point, the poor formulation makes it hard for me to understand all the details. My detailed questions are given in the next section.

[ref1] Sundberg, Patrik, et al. "Occlusion boundary detection and figure/ground assignment from optical flow." CVPR 2011

[ref2] Lu, Yao, et al. "Salient object detection using concavity context." ICCV 2011.

---

> ### Author Response · Authors · 2022-08-02
> **To Reviewer ghhU (1/3)**
>
> **Q#:** The core idea and motivation of our paper.
>
> **A#:** Before answering other questions, we would like to clarify this paper's core idea and motivation. Our work aims to explore the F-G assignment mechanism, which conforms to human vision cognitive theory, to empower CNN to achieve a robust perceptual organization despite visual ambiguity. Existing approaches built on CNNs contain two main components: encoders and contextual modeling. The encoding part typically utilizes a feature extractor (e.g., Resnet) pre-trained on a large-scale classification task (ImageNet). The advantage of doing so is using a large-scale classification task to drive the model to learn associations between pixels and category labels, which can extract invariant representations beyond the pixel level. Nevertheless, such invariant representations lack spatial contextual awareness, which is not friendly for pixel-level prediction tasks like image segmentation. To address this problem, a series of works have tried to complement it by adding or enhancing the capabilities of contextual modeling (multi-scale, multi-receptive-field, long-range modeling, Etc.). However, the visual ambiguities impede CNN's represent learning and contextual modeling, leading to inaccurate and incomplete perceptual organization. Since the visual appearance differences between the foreground and background are obscure, it is difficult to perceive the correlation between individual visual elements and determine the boundaries.
>
> The Figure-Ground assignment principle in human cognition is thought to be important in reducing the visual ambiguousness of a scene from two aspects: 1) as stated in Line 68-71, "neural activity associated with the Figure-Ground assignment mechanism in the V2 cortex of human vision occurs as early as 10-25 ms after the generation of visual stimuli, providing strong support for the role of local bottom-up processing". Equally important, 2) as stated in Line 71-73, "the 'meaningfulness principle' also showed that assigned figures tend to be associated with neighborhoods with familiar shapes, pointing to the integration of knowledge from the top-down".
>
> Inspired by these studies, we correspondingly propose our insights: 1) we perform a pre-attentive selection of supervisory signals (figure-ground cues) and filter out the knowledge that is valuable for context modeling as support (stronger supervisory signals) for the following perception. 2) we introduce a progressive interaction enhancement mechanism (implemented by several IEMs) to support the boundary assignment process.
>
> **Q#:** The figure-ground cues selection.
>
> **A#:** We carefully investigate the configural cues related to the Figure-Ground assignment mechanism in human psychophysics and find that the figural region usually takes on the shape instructed by the separating boundary and appears closer to the viewer. In contrast, the ground region is seen as extending behind the figure. Therefore, we selected cues (Convexity and Lower Region) directly related to the implementation of the figure-ground assignment mechanism in our work. As shown in Line 81, "Convexity cue corresponds to the regions on either side of the boundary where the scene depth changes abruptly and is beneficial for analyzing the hierarchical relationship between neighboring regions in the image, facilitating hierarchical contextual modeling." As shown in Line 84, "Lower region cue usually corresponds to the region of the scene where occlusion has occurred and is beneficial for analyzing the occlusion relationship between various neighborhoods in an image and determining the shape attribution of foreground objects and background regions."

---

> > ### Author Response · Authors · 2022-08-02
> > **To Reviewer ghhU (2/3)**
> >
> > **Q#:** The performance on application tasks.
> >
> > **A#:** Our work aims to explore the F-G assignment mechanism to empower CNN to achieve a robust boundary assignment despite visual ambiguity. And we present a novel Figure-Ground-Aided (FGA) module to learn the configural statistics of the visual scene and leverage it for the reduction of visual ambiguity. To clearly discuss its effectiveness, a cognitive visual testing--Figure-Ground Segregation is presented in Section 3. Similar to the control variable method in physical experiments, Figure-Ground Segregation can reliably evaluate the capability of object segmentation models in boundary assignment, excluding irrelevant factors and some short-cut cues (e.g., context information). The comprehensive experiments adequately demonstrate that our proposed FGA module can facilitate the CNN to learn more efficiently in the reduction of visual ambiguities with low data requirements.
> >
> > Furthermore, the application tasks serve as validation of the potential of our approach. Experimental results show that we achieve superior results in most metrics for all four applications (COD, PS, LIS, and SOD), and we believe that careful tuning of the parameters will result in better performance. To demonstrate the potential of our method, we have optimized some hyperparameters (learning rate and decay strategy) based on the original model, and the experiments show that the performance of our method can be further improved. Where "Our*" represents the performance after optimizing the hyperparameters.
> >
> > The results on CHA Dataset.
> > |            |  S | E      | F      | M      |
> > |:--------:|:--------:|:--------:|:--------:|:--------:|
> > |   [10]   |  0.888   | 0.942  | 0.816  | 0.030  |
> > |   [67]   |  0.888   | 0.945  | 0.796  | 0.031  |
> > |    Our      |  0.898   | 0.945  | 0.839  | 0.032  |
> > |    Our*     |  0.902   | 0.949  | 0.840  | 0.031  |
> >
> > The results on CAM Dataset.
> > |             |  S       | E      | F      | M      |
> > |:--------:|:--------:|:--------:|:--------:|:--------:|
> > |   [10]   |  0.820   | 0.882  | 0.743  | 0.070  |
> > |   [67]   |  0.785   | 0.859  | 0.686  | 0.086  |
> > |    Our      |  0.793   | 0.858  | 0.745  | 0.078  |
> > |    Our*     |  0.803   | 0.871  | 0.748  | 0.068  |
> >
> > The results on COD Dataset.
> > |             |  S       | E      | F      | M      |
> > |:--------:|:--------:|:--------:|:--------:|:--------:|
> > |   [10]   |  0.815   | 0.887  | 0.680  | 0.037  |
> > |   [67]   |  0.818   | 0.850  | 0.667  | 0.035  |
> > |    Our      |  0.818   | 0.888  | 0.683  | 0.034  |
> > |    Our*     |  0.821   | 0.895  | 0.687  | 0.031  |
> >
> > The results on COVID-19 Dataset.
> > |             |  Dice       | Sen      | Spec      | S      | E      | M      |
> > |:--------:|:--------:|:--------:|:--------:|:--------:|:--------:|:--------:|
> > |   [13]   |  0.682   | 0.692  | 0.943  | 0.781  | 0.720  | 0.082  |
> > |   [20]   |  0.700   | 0.751  | -  | -  | 0.860  | 0.084  |
> > |    Our      |  0.735   | 0.720  | 0.965  | 0.792  | 0.900  | 0.062  |
> > |    Our*     |  0.754   | 0.748  | 0.973  | 0.799  | 0.911  | 0.056  |
> >
> > [10] D.-P. Fan, G.-P. Ji, M.-M. Cheng, and L. Shao. Concealed object detection. T-PAMI, 2021.
> >
> > [13] D.-P. Fan, T. Zhou, G.-P. Ji, Y. Zhou, G. Chen, H. Fu, J. Shen, and L. Shao. Inf-net: Automatic covid-19 lung infection segmentation from ct images. T-MI, 2020.
> >
> > [20] G.-P. Ji, L. Zhu, M. Zhuge, and K. Fu. Fast camouflaged object detection via edge-based reversible re-calibration network. PR, 2021.
> >
> > [67] F. Yang, Q. Zhai, X. Li, R. Huang, A. Luo, H. Cheng, and D.-P. Fan. Uncertainty-guided transformer reasoning for camouflaged object detection. CVPR, 2021.
> >
> >
> > **Q#:** I am particularly concerning whether the lower-region cue is well connected to the challenge studied in the work. More discussions are demanded. In addition, I am wondering how will the model perform by directly using a edge-aware loss like HED instead of L_LR.
> >
> > **A#:** Following your comments, we performed an experiment on the normal-level dataset of the Figure-Ground Segregation Test, and the experimental result shows that directly using an edge-aware loss like HED instead of L_LR does not achieve better results than the FGA model itself. The reason is that the supervision using edge-aware only provides the perception of the foreground boundary, while the Lower region cue provides a more contextual perception of the boundary region beyond the foreground boundary. In brief, it allows the model to focus on the associations within the inner and outer regions of the boundary and the differences between the inner and outer regions.
> >
> > |          |    FGA     |    FGA (HED)     |
> > |:--------:|:--------:|:--------:|
> > |     S_a     |    0.816     |    0.807     |
> > |     IoU     |    0.659     |    0.642     |

---

> > > ### Author Response · Authors · 2022-08-02
> > > **To Reviewer ghhU (3/3)**
> > >
> > > **Q#:** The statement "figure-ground assignment ... contributes almost all perception-based tasks" seems an overclaim, or more discussions should be provided to support the argument.
> > >
> > > **A#:** Thanks to your suggestion, we have revised this sentence. Please refer to our rebuttal revision.
> > >
> > > The critical process of the perceptual organization known as figure-ground assignment, first proposed by Edgar Rubin [ref1], involves giving one of the two adjacent regions a boundary. Over the years, many scholars have confirmed its existence and discovered its mechanism. The figure-Ground assignment is commonly thought to follow region segmentation, and it is an essential step in forming a perception of surfaces, shapes, and objects [ref2][ref3].
> > >
> > > [ref1] Rubin, E.: Visuell wahrgenommene figuren. In: Kobenhaven: Glydenalske boghandel. (1921)
> > >
> > > [ref2] J. Wagemans, J. H. Elder, M. Kubovy, S. E. Palmer, M. A. Peterson, M. Singh, and R. von der Heydt. A century of gestalt psychology in visual perception: I. perceptual grouping and figure–ground organization. Psychological bulletin, 2012.
> > >
> > > [ref3] J. Wagemans, J. Feldman, S. Gepshtein, R. Kimchi, J. R. Pomerantz, P. A. Van der Helm, and C. Van Leeuwen. A century of gestalt psychology in visual perception: Ii. conceptual and theoretical foundations. Psychological bulletin, 138(6):1218, 2012.
> > >
> > >
> > > **Q#:** How are K_C and K_L determined? Do they set to the same values across the datasets? How will the different values affect the performance?
> > >
> > > **A#:** The structure element size (K_C and K_L) mentioned in the paper is an intuitive choice. They are set to the same values across the datasets. Table 7-10 show that our proposed method achieves competitive results without hyper-parameter tuning. Furthermore, the physical meaning of K_C and K_L, which control the granularity of the Figure-Ground cues and their coverage areas, is easy to understand. Therefore, we believe carefully selecting this hyperparameter will improve performance.
> > >
> > >
> > > **Q#:** Why is Labmbda used here?
> > >
> > > **A#:** As stated in the caption of Figure 3, we use Lambda strategy, which has been proven to be an effective method, to improve the computation efficiency of the CLI and GI interaction.
> > >
> > >
> > > **Q#:** Typos and other errors.
> > >
> > > **A#:** Thanks for your detailed comments. We have addressed each of these issues. Please refer to our rebuttal revision.

---

> > > > ### Comment · Reviewer_ghhU · 2022-08-08
> > > > **Further comments**
> > > >
> > > > Thanks for the detailed response, which has addressed some of my concerns. Now I have some new comments.
> > > >
> > > > 1) Regarding the selection of two cues, I understand that the selected cues are of importance. However, how about other cues mentioned in the related work section since they are also "factors that affect Figure-Ground assignment".
> > > > 2) For the new results, could you point out which parameters are critical for the performance. Even for these new results, I find that the improvements for some datasets (e.g., CHA, COD) are not significant. While it is not essential to obtain highly superior results, some insights on the results should be provided.
> > > > 3) I am curious about the implementation of HED loss. Did you apply deep supervision, which is known important to the performance?
> > > > 4) The motivation to use Lambda instead of self-attention is not discussed.
> > > > 5) Finally, the writing of the methodology part is not good (as I mentioned in the first-round comments). While some revision has been made, I did not check very carefully (which will take some time) to confirm its correctness.

---

> > > > > ### Author Response · Authors · 2022-08-09
> > > > > **Further Responses**
> > > > >
> > > > > Thanks for your new comments, we clarify your concern below.
> > > > >
> > > > >
> > > > > **Q#:** Regarding the selection of two cues, I understand that the selected cues are of importance. However, how about other cues mentioned in the related work section since they are also "factors that affect Figure-Ground assignment".
> > > > >
> > > > > **A#:** Our work aims to explore the F-G assignment mechanism to empower CNN. A major part of our work is the introduction of FG cues, and we have explained the motivation for choosing two cues in detail in the introduction and in the previous round of responses, based on which we have conducted a follow-up study. We believe that using more valid FG cues can further contribute to developing this direction, which will be an essential part of our future work.
> > > > >
> > > > >
> > > > > **Q#:** For the new results, could you point out which parameters are critical for the performance. Even for these new results, I find that the improvements for some datasets (e.g., CHA, COD) are not significant. While it is not essential to obtain highly superior results, some insights on the results should be provided.
> > > > >
> > > > > **A#:** In our experiments we found that the learning rate is important for performance. By increasing the learning of learnable parameters in FGA module to 1.1 times of the original (the learning rates of Encoder and Decoder are unchanged), the performance can be further improved.
> > > > >
> > > > > In the challenging COD task, each camouflaged image is accompanied by different super-classes (Amphibian, Aquatic, Flying, and Territorial) that reflect common challenges in real-world scenes. These annotations are beneficial for investigating the pros and cons of camouflaged object detection methods. To illustrate the insight of the new results, here we report the results on the Aquatic super-class. This super-class was chosen because most of the images in this superclass contain indefinable boundaries. The result on this superclass set show that the improvement of our method is significant, which further illustrates the effectiveness of FGA.
> > > > >
> > > > > The result on Aquatic (474 images) super-class in COD dataset.
> > > > > |        |  S | E      | F      | M      |
> > > > > |:--------:|:--------:|:--------:|:--------:|:--------:|
> > > > > |   [10]   |  0.811   | 0.883  | 0.696  | 0.051  |
> > > > > |   [67]   |  0.815   | 0.873  | 0.687  | 0.049  |
> > > > > |   Our    |  0.817   | 0.890  | 0.704  | 0.045  |
> > > > > |   Our*   |  0.823   | 0.899  | 0.710  | 0.042  |
> > > > >
> > > > > [10] D.-P. Fan, G.-P. Ji, M.-M. Cheng, and L. Shao. Concealed object detection. T-PAMI, 2021.
> > > > > [67] F. Yang, Q. Zhai, X. Li, R. Huang, A. Luo, H. Cheng, and D.-P. Fan. Uncertainty-guided transformer reasoning for camouflaged object detection. CVPR, 2021.
> > > > >
> > > > >
> > > > > **Q#:** I am curious about the implementation of HED loss. Did you apply deep supervision, which is known important to the performance?
> > > > >
> > > > > **A#:** We directly replaced L_LR with HED loss in the implementation process and did not use the deep supervision strategy. Note that our L_LR also does not use the deep supervision strategy in the implementation process, so we believe the previous experiment is fair. To alleviate your concern, we have also added the experimental results of adding the deep supervision strategy. We also include the deep supervision strategy in the L_LR part for a fair comparison. As we can see from the experimental results, the performance of the FGA (HED) model with deep supervision is further improved and even slightly surpasses that of the original FGA model. However, the performance is still inadequate compared to the FGA model with deep supervision.
> > > > >
> > > > > |          |    FGA     |    FGA +Deep    |    FGA (HED)     |    FGA (HED) +Deep    |
> > > > > |:--------:|:--------:|:--------:|:--------:|:--------:|
> > > > > |     S_a     |    0.816     |    0.832     |    0.807     |    0.815   |
> > > > > |     IoU     |    0.659     |    0.671     |    0.642     |    0.663   |
> > > > >
> > > > >
> > > > > **Q#:** The motivation to use Lambda instead of self-attention is not discussed.
> > > > >
> > > > > **A#:** Lambda layers generalize and extend self-attention formulations to capture both content-based and position-based interactions in global or local, which is crucial for modeling highly structured inputs such as images. Moreover, the modules built on the Lambda strategy are computationally efficient, model long-range dependencies at a small memory cost, and can therefore be well suited for dense prediction tasks. We will incorporate this discussion into the next version.

---

> ### Author Response · Authors · 2022-08-08
> **Request for discussion**
>
> Dear Reviewer ghhU,
>
> We have addressed your major concerns regarding our paper with additional experimental results. We are happy to discuss them with you in the openreview system if you feel that there still are some concerns/questions. We also welcome new suggestions/comments from you!
>
> Best regards,

---

### Official Review · Reviewer_dfqS · 2022-07-12

**Rating:** 5
**Confidence:** 4
**Soundness:** 3 good
**Presentation:** 2 fair
**Contribution:** 3 good

**Summary:**

The paper proposes a module for enhancing figure-ground assignment, and which can be used to extend existing encoder-decoder based segmentation models. More specifically, the authors implement mechanisms for measuring two figure-ground cues, namely ‘convexity’ and ‘lower region’. These two cues come from the cognitive vision literature, implemented via morphological operators, and inserted into the encode-decoder CNN architecture. Experimental evaluation of known CNNs augmented with this novel module are then performed on challenging segmentation tasks, e.g., those of medical imaging, indicating an improvement of accuracy in most tasks.

**Questions:**

The IEM is built upon adding more learning units through the spatial self-attention mechanism. I wonder if adding parameters, and/or depth can improve the IoU and S-measure on the FG test set. Can you comment on the number of parameters added by the IEM units to the ResNet50, and speculate about performance of deeper segmentation models?

**Limitations:**

It will be good to exemplify few failure cases of your model (e.g., on the FG or medical datasets). Perhaps other FG factors are needed? (e.g., good continuation?).

**Strengths And Weaknesses:**

Strengths:
•	Figure-ground assignment is indeed a fundamental process for computer vision algorithms, which is yet not fully explored. I like the idea of adding FG mechanisms to the current FCN segmentation models.
•	The FG segregation test is an important test, which is indeed popular on the human vision literature. I agree that segmentation models such as DeeplabV3 should be able to replicate human behavior in this test set, with and without additional training from a similar set.

Weaknesses:
•	The paper is a bit hard to follow, and several sections were needed more than one reading pass. I suggest improving the structure (introduction->method->experiments), and put more focus on the IEM in Fig 3, which is in my view the main figure in this paper. Also, to improve the visualization of the Fig 7, and Fig. 10.

---

> ### Author Response · Authors · 2022-08-02
> **To Reviewer dfqS**
>
> Thank you for your feedback. We hope you will agree that we have addressed your main concerns, and that the paper is far more readable because of it.
>
> **Q#:** I suggest improving the structure (introduction->method->experiments), and put more focus on the IEM in Fig 3, which is in my view the main figure in this paper. Also, to improve the visualization of the Fig 7, and Fig. 10.
>
> **A#:** Thanks to your suggestion, we have modified the structure of our paper. Please refer to our rebuttal revision.
>
> We need to clarify that our work aims to explore the F-G assignment mechanism, which conforms to human vision cognitive theory, to empower CNN to achieve a robust perceptual organization despite visual ambiguity. Specifically, we present a novel Figure-Ground-Aided (FGA) framework to learn the configural statistics of the visual scene and leverage it for the reduction of visual ambiguity. Two main aspects are involved in our method: 1) We perform a pre-attentive selection of supervisory signals (figure-ground cues), and filter out the knowledge that is valuable for context modeling as support (stronger supervisory signals) for the following perception; 2) We introduce a progressive interaction enhancement mechanism (implemented by several IEMs) to provide support for the boundary assignment process. Both of these contributions are equally important for our work.
>
> In addition, due to the page limit of the paper, we provide the visualization of Table 7-10 in the Supplementary Material, as detailed in Figure S8, Figure S10, and Figure S11.
>
>
> **Q#:** The IEM is built upon adding more learning units through the spatial self-attention mechanism. I wonder if adding parameters, and/or depth can improve the IoU and S-measure on the FG test set. Can you comment on the number of parameters added by the IEM units to the ResNet50, and speculate about performance of deeper segmentation models?
>
> **A#:** Following your comments, we explored the relevant hyperparameters of IEM on the normal-level dataset of the Figure-Ground Segregation Test. We explored the relative positional embedding range of the CLI part of the IEM, a parameter that controls the scale of the IEM for modeling local contexts. As shown in the table below, the performance improves further as the relative position embedding range increases, but the magnitude of the improvement becomes diminished. Moreover, to further reduce the number of parameters and computation in IEM, we incorporate a bottleneck strategy similar to that in ResBlock, where we explore the impact of the scaling ratio of bottlenecks in IEM on performance. Experimental results show that the smaller the scaling ratio (indicating a larger number of parameters), the performance can be improved, but the improvement is limited. Details of the python implementation of IEM are provided in the supplementary material (Figure S2).
>
> The relative positional embedding range (r) w.r.t performance.
> |     r     |    3     |    5     |     7    |     9    |
> |:--------:|:--------:|:--------:|:--------:|:--------:|
> |     S_a     |    0.813     |    0.816     |     0.818    |     0.818    |
> |     IoU     |    0.690     |    0.695     |     0.696    |     0.698    |
>
> The scaling ratio (rate) of bottlenecks in IEM w.r.t performance.
> |     rate     |    32     |    16     |     8    |
> |:--------:|:--------:|:--------:|:--------:|
> |     S_a     |    0.816     |    0.819     |     0.819    |
> |     IoU     |    0.695     |    0.700     |     0.703    |
>
> The number of parameters added by the IEM units to the ResNet50 is 7.97M, when the relative position embedding range in the CLI is 5, and the compression ratio of the bottleneck is 32. Moreover, a deeper segmentation model means it has a better representation capability, and building upon a better representation, we believe our approach can achieve better performance.
>
>
> **Q#:** Failure case of our method.
>
> **A#:** Thanks for your suggestion. We have presented the failure case in the supplementary material. As shown in Figure S12, our method produces an inaccurate result in the case of excessively complex foreground content. This is probably due to the fact that our FGA model learns configural statistics that do not contain these local trivial details. To cope with this situation, we can add specific network [ref1] or post-processing measures [ref2] used to refine the details behind the existing model.
>
> [ref1] Qin, X., Zhang, Z., Huang, C., Gao, C., Dehghan, M. and Jagersand, M., 2019. Basnet: Boundary-aware salient object detection. CVPR.
>
> [ref2] Krähenbühl, P. and Koltun, V., 2011. Efficient inference in fully connected crfs with gaussian edge potentials. Advances in neural information processing systems.

---

### Official Review · Reviewer_fHq8 · 2022-07-12

**Rating:** 5
**Confidence:** 4
**Soundness:** 3 good
**Presentation:** 3 good
**Contribution:** 2 fair

**Summary:**

This paper focuses on perceptual organizatio where the goal is to perceive and group the individual visual element. To address the visual ambiguity in the discrimination process, authors propose to use e figure-ground assignment mechanism to improve deep neural networks  to achieve a robust perceptual organization. Specifically, a Figure-Ground-Aided (FGA) module is presented to learn the configural statistics of the visual scene and leverage it for the reduction of visual ambiguity. An Interactive Enhancement Module (IEM) is designed to leverage configural cues from FGA to enhance the representation learning. Experiments are conducted on four applications.

**Questions:**

Please refer to the weakness part in theStrengths And Weaknesses section.

**Limitations:**

Authors discussed the limitations of the method in the main paper and the appendix.

**Strengths And Weaknesses:**

Strength:

+ The idea of taking the inspiration from the perceptual organization of human vision and studying the figure-Ground cues is interesting. The motivation is also well explained.

+ Experiments are conducted on different robust perception organization tasks, such as camouflaged object detection, polyp segmentation, and lung infection segmentationm. The results validate that the proposed method can help improve the performance of robust perception organization tasks.

+ The proposed Figure-Ground Segregation test can help to investigate the performance of models in Figure-Ground assignment.


Weakness:

- The performance gaps between the proposed method and existing methods are marginal in some metrics. The best numbers are not always marked in bold which is missleading, e.g. second row of "Figure 9", second row of "Figure 10".

- The main experimental result tables are titled as Figures, such as Figure7, 8, 9, 10. To me, they are tables rather than fingures.

---

> ### Author Response · Authors · 2022-08-02
> **To Reviewer fHq8**
>
> Thank you for the detailed comments. These comments would help us greatly improve the quality of the paper.
>
> **Q#:** The performance gaps between the proposed method and existing methods are marginal in some metrics.
>
> **A#:** Our work aims to explore the F-G assignment mechanism to empower CNN to achieve a robust boundary assignment despite visual ambiguity. And we present a novel Figure-Ground-Aided (FGA) module to learn the configural statistics of the visual scene and leverage it for the reduction of visual ambiguity. To clearly discuss its effectiveness, a cognitive visual testing--Figure-Ground Segregation is presented in Section 3. Similar to the control variable method in physical experiments, Figure-Ground Segregation can reliably evaluate the capability of object segmentation models in boundary assignment, excluding irrelevant factors and some short-cut cues (e.g., context information). The comprehensive experiments adequately demonstrate that our proposed FGA module can facilitate the CNN to learn more efficiently in the reduction of visual ambiguities with low data requirements.
>
> Furthermore, the application tasks serve as validation of the potential of our approach. Experimental results show that we achieve superior results in most metrics for all four applications (COD, PS, LIS, and SOD), and we believe that careful tuning of the parameters will result in better performance. To demonstrate the potential of our method, we have optimized some hyperparameters (learning rate and decay strategy) based on the original model, and the experiments show that the performance of our method can be further improved. Where "Our*" represents the performance after optimizing the hyperparameters.
>
> The results on CHA Dataset.
> |            |  S | E      | F      | M      |
> |:--------:|:--------:|:--------:|:--------:|:--------:|
> |   [10]   |  0.888   | 0.942  | 0.816  | 0.030  |
> |   [67]   |  0.888   | 0.945  | 0.796  | 0.031  |
> |    Our      |  0.898   | 0.945  | 0.839  | 0.032  |
> |    Our*     |  0.902   | 0.949  | 0.840  | 0.031  |
>
> The results on CAM Dataset.
> |             |  S       | E      | F      | M      |
> |:--------:|:--------:|:--------:|:--------:|:--------:|
> |   [10]   |  0.820   | 0.882  | 0.743  | 0.070  |
> |   [67]   |  0.785   | 0.859  | 0.686  | 0.086  |
> |    Our      |  0.793   | 0.858  | 0.745  | 0.078  |
> |    Our*     |  0.803   | 0.871  | 0.748  | 0.068  |
>
> The results on COD Dataset.
> |             |  S       | E      | F      | M      |
> |:--------:|:--------:|:--------:|:--------:|:--------:|
> |   [10]   |  0.815   | 0.887  | 0.680  | 0.037  |
> |   [67]   |  0.818   | 0.850  | 0.667  | 0.035  |
> |    Our      |  0.818   | 0.888  | 0.683  | 0.034  |
> |    Our*     |  0.821   | 0.895  | 0.687  | 0.031  |
>
> The results on COVID-19 Dataset.
> |             |  Dice       | Sen      | Spec      | S      | E      | M      |
> |:--------:|:--------:|:--------:|:--------:|:--------:|:--------:|:--------:|
> |   [13]   |  0.682   | 0.692  | 0.943  | 0.781  | 0.720  | 0.082  |
> |   [20]   |  0.700   | 0.751  | -  | -  | 0.860  | 0.084  |
> |    Our      |  0.735   | 0.720  | 0.965  | 0.792  | 0.900  | 0.062  |
> |    Our*     |  0.754   | 0.748  | 0.973  | 0.799  | 0.911  | 0.056  |
>
> [10] D.-P. Fan, G.-P. Ji, M.-M. Cheng, and L. Shao. Concealed object detection. IEEE transactions on pattern analysis and machine intelligence, 2021.
>
> [13] D.-P. Fan, T. Zhou, G.-P. Ji, Y. Zhou, G. Chen, H. Fu, J. Shen, and L. Shao. Inf-net: Automatic covid-19 lung infection segmentation from ct images. IEEE transactions on medical imaging, 2020.
>
> [20] G.-P. Ji, L. Zhu, M. Zhuge, and K. Fu. Fast camouflaged object detection via edge-based reversible re-calibration network. Pattern Recognition, page 108414, 2021.
>
> [67] F. Yang, Q. Zhai, X. Li, R. Huang, A. Luo, H. Cheng, and D.-P. Fan. Uncertainty-guided transformer reasoning for camouflaged object detection. In Proceedings of the IEEE/CVF International Conference on Computer Vision, pages 4146–4155, 2021.
>
>
> **Q#:** The best numbers are not always marked in bold which is missleading, e.g. second row of "Figure 9", second row of "Figure 10". The main experimental result tables are titled as Figures, such as Figure7, 8, 9, 10. To me, they are tables rather than fingures.
>
> **A#:** As the reviewer pointed out, there are some annotation errors in "Figure 9" and "Figure 10". We have checked and corrected them in the rebuttal version. Furthermore, we have modified the title of the experimental results section from "Figure" to "Table". Please refer to our rebuttal revision.

---

### Author Response · Authors · 2022-08-02
**Thanks for the reviews**

We sincerely thank the reviewers for their thoughtful reviews. We have provided detailed responses in individual responses to each reviewer and incorporated feedback in the revised version. We believe that these revisions will greatly improve the manuscript. You can find the version of the manuscript by clicking on the "Show revisions" link. The link contains a revised manuscript and supplementary material.

---

### Meta-Review · Area_Chair_TcUH · 2022-08-21

**Recommendation:** Accept
**Confidence:** Certain

**Metareview:**

Overall, the reviewers commend the motivation of the approach, the core ideas presented in the paper, and the extensive experiments conducted for four different applications including camouflaged and salient object detection, infection, and polyp segmentation.

In response to Reviewer fHq8, the authors have mentioned updated results with hyper-parameter tuning, however, they don’t mention which set is used for this purpose. Details on whether the validation set is used or not and how it is chosen are important for the final version.

In response to Reviewer ghhU, authors have reported new experiments and comparisons, alongside clarifications on motivation and justification for the choices made as part of the approach.

It appears that the major concerns from reviewers have been addressed in the response and the paper can be accepted after the rebuttal. Authors are suggested to include all the suggested changes in the final version.

**Award:**

No

---

### Decision · Program_Chairs · 2022-09-14

Accept